Virus-derived sequences from the transcriptomes of two snail vectors of schistosomiasis, Biomphalaria pfeifferi and Bulinus globosus from Kenya

Liu Sijun 1
Zhang Si-Ming 2
http://orcid.org/0000-0002-9506-9259 Buddenborg Sarah K. 3
http://orcid.org/0000-0002-2916-9868 Loker Eric S. 3
http://orcid.org/0000-0002-9956-9613 Bonning Bryony C. 4 bbonning@ufl.edu
1 Department of Entomology, Iowa State University , Ames, Iowa , United States
2 Center for Evolutionary and Theoretical Immunology, Parasite Division Museum of Southwestern Biology, Department of Biology, University of New Mexico , Albuquerque, New Mexico , United States
3 Department of Biology, University of New Mexico , Albuquerque, New Mexico , United States
4 Entomology & Nematology Department, University of Florida , Gainesville, Florida , United States
Breitbart Mya
Electronic publication date: 2021 Nov 15
Publication date: 2021
Volume: 9
Electronic Location ID: e12290
Received 2020 Apr 8; Accepted 2021 Sep 21
Copyright: © 2021 Liu et al.
Copyright year: 2021
Copyright holder: Liu et al.
License: This is an open access article distributed under the terms of the Creative Commons Attribution License, which permits unrestricted use, distribution, reproduction and adaptation in any medium and for any purpose provided that it is properly attributed. For attribution, the original author(s), title, publication source (PeerJ) and either DOI or URL of the article must be cited.
License URL: https://creativecommons.org/licenses/by/4.0/

Keywords: Snail virus, Schistosomiasis vector, Virus discovery, Biomphalaria pfeifferi, Bulinus globosus, Kenya, RNA virus, Diatom/marine virus, Iflavirus, Dicistrovirus

Funding: National Institute of General Medical Sciences of the National Institutes of Health P30GM110907 National Institute of Health (NIH) R37AI101438 This work was supported by the National Institute of General Medical Sciences of the National Institutes of Health under Award Number P30GM110907 and the National Institute of Health (NIH) grant R37AI101438. The funders had no role in study design, data collection and analysis, decision to publish, or preparation of the manuscript.

==============================
Schistosomiasis, which infects more than 230 million people, is vectored by freshwater snails. We identified viral sequences in the transcriptomes of Biomphalaria pfeifferi (BP) and Bulinus globosus (BuG), two of the world’s most important schistosomiasis vectors in Africa. Sequences from 26 snails generated using Illumina Hi-Seq or 454 sequencing were assembled using Trinity and CAP3 and putative virus sequences were identified using a bioinformatics pipeline. Phylogenetic analyses were performed using viral RNA-dependent RNA polymerase and coat protein sequences to establish relatedness between virus sequences identified and those of known viruses. Viral sequences were identified from the entire snail holobiont, including symbionts, ingested material and organisms passively associated with the snails. Sequences derived from more than 17 different viruses were found including five near full-length genomes, most of which were small RNA viruses with positive sense RNA genomes (i.e., picorna-like viruses) and some of which are likely derived from adherent or ingested diatoms. Based on phylogenetic analysis, five of these viruses (including BPV2 and BuGV2) along with four Biomphalaria glabrata viruses reported previously, cluster with known invertebrate viruses and are putative viruses of snails. The presence of RNA sequences derived from four of these novel viruses in samples was confirmed. Identification of the genome sequences of candidate snail viruses provides a first step toward characterization of additional gastropod viruses, including from species of biomedical significance.

Introduction

Several species of freshwater snail play an integral role in transmission of schistosomiasis, a tropical disease that infects over 230 million people with a further 780 million at risk, primarily in sub-Saharan Africa (Colley et al., 2014). In Africa, snails of the genera Biomphalaria and Bulinus transmit human intestinal and urinary schistosomiasis, respectively. Snails serve as the obligatory hosts of the asexually-reproducing larval stages of schistosomes that can produce hundreds to thousands of human-infective cercariae per day. Some infected snails can continue to release cercariae into the water for several months (Mutuku et al., 2014), posing a significant hurdle for those hoping to prevent new schistosome infections in people.

Biological control provides an alternative approach to target the snail hosts of schistosomiasis, and both snail predators (prawns, Macrobrachium) (Sokolow et al., 2015), and snail pathogens (bacteria, Paenibacillus) (Duval et al., 2015a; Duval et al., 2015b) have potential in this regard. As viruses also have potential for use in the biological control of snails similar to other pestiferous invertebrates (Sosa-Gómez et al., 2020), we sought to characterize the virome of two snail vector species.

In the search for snail viruses, the presence of contaminating viruses from the surrounding environment must be taken into consideration. Viral sequences discovered from seawater (Culley, Lang & Suttle, 2003; Culley, Lang & Suttle, 2006; Culley, Lang & Suttle, 2007; Culley et al., 2014; Culley & Steward, 2007; Lopez-Bueno et al., 2015), freshwater (Bolduc et al., 2012; Djikeng et al., 2009; Mohiuddin & Schellhorn, 2015) and aquatic unicellular organisms (Kimura & Tomaru, 2015; Lang, Culley & Suttle, 2004; Nagasaki et al., 2004; Nagasaki et al., 2005; Takao et al., 2006) have revealed that the vast majority of viruses in aquatic environments are small RNA viruses with positive sense, single stranded RNA genomes (SRV+) similar to the eukaryotic virome (Culley, Lang & Suttle, 2003; Suttle, 2005; Suttle, 2007). These RNA viruses show genomic, structural and protein sequence similarities to insect viruses, including dicistroviruses and iflaviruses (Suttle, 2007). However, the majority of the SRV+ discovered recently from aquatic environments are not well characterized with only a few SRV+ that infect unicellular organisms having been studied (Kimura & Tomaru, 2015; Lang, Culley & Suttle, 2004; Nagasaki et al., 2004; Nagasaki et al., 2005). SRV+ have also been isolated from shrimp (Mari et al., 2002), crabs (Boros et al., 2011), bivalves (Rosani & Gerdol, 2017; Rosani et al., 2019) and partial sequences of virus-derived genes identified from the crop of an invasive snail (Cardoso et al., 2012). Several novel SRV+ from Biomphalaria glabrata were isolated by analyzing available RNA sequence data for this Neotropical snail (Adema et al., 2017; Galinier et al., 2017), and analysis of the transcriptome of B. pfeifferi revealed nearly 1,300 transcripts putatively derived from viruses (Buddenborg et al., 2017).

Here we report use of our bioinformatics pipeline (Liu, Vijayendran & Bonning, 2011), to identify viral sequences in large sequence databases derived from two of the most important schistosomiasis snail vectors in Africa (Biomphalaria pfeifferi and Bulinus globosus) and examined the RNA sequences for evidence of the presence of viruses. We identified sequences derived from multiple viruses that based on phylogenetic analysis, are predicted to infect and replicate in the snail. We also identified sequences from numerous other viruses including four near full length genomes, that are likely infectious to diatoms and other aquatic microorganisms associated with the snail.

Materials and Methods

Collection of snails

Individual snails of two species, Biomphalaria pfeifferi (BP) and Bulinus globosus (BuG), were collected from Kenya. BP snails (1–21, Table 1) were collected from Kasabong Stream, West Kenya (00°9″6.84″S, 34°20′7.80″E) in 2013 and treated as described previously to test snail responses to schistosome infection (Buddenborg et al., 2017). The sample set includes control snails (uninfected with schistosomes) and snails representing various infection treatments. BuG samples (22–24, Table 1) were collected from Mombasa, East Kenya (00°3′54.83″S, 39°31.72″E) in 2013. Two additional BP snails (25–26, Table 1) were collected from Asao Stream, West Kenya (00°19′5.50″S, 35°0′24.99″E) in 2012. Snails were placed individually into 1.5 ml tubes with one ml of TRIzol (Invitrogen, Carlsbad, CA, USA) and stored at −80 °C until RNA extraction. This project was undertaken with approval of Kenya’s National Commission for Science, Technology and Innovation (permit number NACOSTI/P/15/9609/4270), Kenya Wildlife Service (KWS-0045-03-21 and export permit 0004754), and National Environment Management Authority (NEMA/AGR/46/2014 and NEMA/AGR/149/2021).

Table 1 Snail samples from which virus sequences were derived.

Species	Sample	Abbreviation (Buddenborg et al., 2017)	Sequencing method	No. virus-derived contigs	
Biomphalaria pfeifferi
(Kasabong)	1–3	Control-R1 to R3	Illumina RNA-Seq	50	
4–6	Shedding R1-R3	43	
7–9	1d-R1 to R3	8	
10–12	3d-R1 to R3	21	
13–15	BP control × molluscicide	30	
16–18	BP infected × molluscicide	45	
19–21	–	9	
Bulinus globosus
(Mombasa)	22–24	–	9	
B. pfeifferi
(Asao)	25	–	454 GS FLX	0	
26	–	17	

Extraction of RNA, library preparation and sequencing

Methods for RNA extraction from snail samples (Table 1) and high-through put sequencing were as described previously (Buddenborg et al., 2017). Briefly, snails were homogenized individually in Trizol using plastic pestles (USA Scientific, Ocala, FL, USA), followed by RNA extraction according to the manufacturer’s instructions (Invitrogen, Carlsbad, CA, USA). Genomic DNA contamination was removed with RNase-free DNase I (New England BioLabs, Ipswich, MA, USA) at 37 °C for 10 min. Samples were further purified using the PureLink RNA Mini Kit (Thermo Scientific, Waltham, MA, USA). RNA quality and quantity were evaluated on a Bioanalyzer 2100 (Agilent Technologies, Santa Clara, CA, USA) and Nanodrop ND-1000 (Thermo Scientific, Waltham, MA, USA). Complementary DNA was prepared and Illumina Hi-Seq sequencing conducted at the National Center for Genomic Research in Santa Fe, New Mexico (http://www.ncgr.org). Complementary DNA libraries were paired-end sequenced (2 × 50 base reads) on a HiSeq2000 instrument (Illumina, Carlsbad, CA, USA). A total of 20 to 30 million reads of 50 nucleotides were generated from each individual snail with three sequencing replicates per sample (Buddenborg et al., 2017). Sequence data are available at NCBI under BioProject ID PRJNA383396. RNA-Seq data were also acquired for BP samples 25 and 26 (Table 1) on the 454 GS FLX platform (Roche, Basel Switzerland) at the University of New Mexico Biology Molecular Biology Facility using standard procedures (http://ceti.unm.edu).

The quality of sequencing reads was examined using FastQC v0.11.3 (Andrews http://www.bioinformatics.babraham.ac.uk/projects/fastqc/). When necessary, bases with low quality scores (quality score threshold, q = 30) were trimmed using Fastx-toolkit v0.0.13 (http://hannonlab.cshl.edu/fastx_toolkit/). Viral genome-derived sequences were either assembled from sequence data from individual snails, or from the combined sequence data from three replicate snails (see sample lane, Table 1). The Illumina sequencing reads and 454 sequencing data were assembled using the Trinity assembler (v2.0.6) (Haas et al., 2013). The contigs obtained from the Trinity assembly were then further assembled by using the CAP3 assembler (Huang & Madan, 1999) to merge redundant sequences. The merged contigs (≥300 nt) were used for annotation to identify viral sequences. Virus-derived contigs that encoded >800 aa and represented sequences derived from novel viruses were selected for analysis of viral sequence abundance. To determine relative abundance of the sequencing reads derived from viral sequences, MAQ v0.5.0 (http://maq.sourceforge.net/maq-man.shtml; Command maq map using default settings) was used for mapping of reads to target assembled sequences.

Identification and annotation of viral sequences

To identify putative viral sequences, the assembled contigs were annotated by BLAST search v.2.10.0 (Camacho et al., 2009) against a local viral protein database of viral proteins from RefSeq (Release 93). Contigs that hit viral proteins with a cutoff E-Value of 0.01 were extracted and further annotated by in-house BLAST against the NCBI nr database. The contigs with viral hits were extracted and sequences derived from putative viruses further analyzed: Contigs that hit the same viral group were manually checked using BioEdit v7.2.5 for potential generation of longer sequence fragments.

Confirmation of viral sequences

The presence of assembled viral sequences was confirmed for BPV2, BPV3, BPV4 and BuGV1 by RT-PCR. Complementary DNA was generated from the snail-derived RNA used for sequencing, by using the SuperScript III First-Strand Synthesis System for RT-PCR (Invitrogen, Waltham, MA, USA) according to the manufacturer’s directions. Primer design was based on the assembled viral sequences (Table S1). PCR was conducted using GoTaq DNA polymerase (Promega, Madison, WI, USA). For short read (<1.5 kb) amplification, PCR was conducted with an initial 94 °C for 5 min, followed by 94 °C, 1 min, 62 °C, 1 min, 72 °C, 5 min for 30 cycles, and 72 °C for 10 min. For long read (>1.5 kb) amplification, PCR was conducted with an initial 95 °C for 5 min, followed by 95 °C, 45 s, 56 °C for 45 s, 72 °C 2 min for 35 cycles, and 72 °C for 8 min.

Multiple sequence alignment, conserved domain identification and phylogenetic analysis

Conserved domains of the viral sequences identified were used for phylogenetic analysis. The conserved domains of RNA-dependent RNA polymerase (RdRP) and capsid proteins or putative capsid sequences were identified by BLAST domain search against the NCBI nr database (Marchler-Bauer et al., 2017). Internal ribosome entry sites (IRES) of dicistro-like viruses were predicted by IRESite (http://iresite.org/). Sequences of conserved domains were extracted and used for phylogenetic analysis. The domain sequence was aligned using MAFFT (with MAFFT-auto and default parameters) (Katoh & Standley, 2013). Phylogenetic trees were constructed and generated with IQ-TREE using maximum-likelihood (Nguyen et al., 2015) with 10,000 bootstraps (Hoang et al., 2018). The best partitioning scheme and models were selected by ModelFinder (Kalyaanamoorthy et al., 2017). The phylogenetic trees were also drawn using FigTree (v1.4.2) (Notredame, Higgins & Heringa, 2000). General sequence manipulations (editing and alignment) were performed using BioEdit v7.2.5 (http://www.mbio.ncsu.edu/BioEdit/page2.html).

Results

Viral sequences in the snail transcriptome

The number of contigs derived from putative viral sequences from the snail samples is shown in Table 1. As our primary targets were sequences derived from actual viruses rather than viral sequences integrated into the genome of the snail (primarily retrotransposon sequences), putative endogenous viral element (EVE) sequences identified by BLAST were not included in this analysis. Redundant sequences were removed from putative viral contigs. Five near full-length SRV+ genome sequences (four from BP and one from BuG; Table 2) were assembled from the remaining contigs. In addition, 54 and three non-redundant partial viral genome sequences were also identified from BP and BuG, respectively (Table S2). The vast majority of the viral contigs were derived from SRV+ with most top hits to viruses from aquatic environments, particularly to unicellular photoautotrophs. Other sequences hit insect SRV+, demonstrating considerable diversity in the populations of viruses associated with the snails.

Table 2 Distribution of virus-derived sequences among snail samples.

Sp.	Sample	BPV1a, b	BPV2a	BPV3a	BPV4a	BPVS1	BPVS6	BPVS16	BPVS17	BPVS33	BPVS42	BuGV1a	BuGV2	
BP	1–3	0	44,740/263c	5,781/35	1,495/9	900	863	278	903	42	817	0	0	
4–6	0	12,305/72	2,713/16	1,670/10	463	1,282	1,666	642	62	123	0	0	
7–9	0	5,468/32	926/6	2,544/15	223	74	216	265	5	39	0	0	
10–12	0	2,107/12	571/3	175/1	189	72	30	203	793	19	0	0	
13–15	0	28,951/170	2,709/16	9,222/56	1,202	744	426	1,181	113	1,452	0	0	
16–18	0	19,084/112	2,008/12	5,792/35	607	2,725	740	781	210	209	0	0	
19–21	0	492/6 (#21)	218/3	0	0	5,150	0	0	667	55	0	0	
BuG	22–24	0	0	0	0	0	3	0	0	0	0	24,622/298	134/15	
Notes:

Sample designations are as described in Table 1.

The distribution of virus-derived sequences is shown for five near-complete and seven partial viral genomes. Sequencing reads were mapped to assembled viral sequences using the MAQ assembly program.

a Near full-length genome sequence identified.

b Isolated from BP454 sequencing data only (not shown here).

c Number of reads mapped/fold-coverage of the assembled genome.

Viral sequences identified from B. pfeifferi

A novel small RNA virus discovered from 454 sequencing data

For the two BP snails from Asao, Kenya, 17 contigs derived from putative viral sequences were identified from approximately 3,000 contigs (>300 nt) assembled by Trinity. One near complete small RNA viral genome (named Biomphalaria pfeifferi-associated virus 1, BPV1: GenBank accession number MT108488; Text S1) was recovered. BPV1 appeared to be a novel dicistro-like virus of at least 8,367 nt (Fig. 1). The predicted ORF1 polyprotein of BPV1 showed 30% protein sequence identity with the counterpart predicted polyprotein of Marine RNA virus JP-A (MRNAVJPA) (Culley, Lang & Suttle, 2007), a Marnavirus of Marnaviridae in the order Picornavirales (http://ictvonline.org/virusTaxonomy.asp). The second closest hit was Rhizosolenia setigera RNA virus 01 (RhsRNAV01) (Nagasaki et al., 2004), with 27% protein sequence identify. RhsRNAV01 is the type member of a recently established genus, Bacillarnavirus of Picornavirales. Two conserved domains, RNA-helicase and RdRP (pfam00910, cd01699 and pfam00680) of dicistroviruses were identified in ORF1 (Fig. 1). ORF2 encodes putative coat proteins (CP). Four conserved CP domains of SRV+ were identified: two Rhv-like CP domains (CP, cd00205, pfm00073), dicistro_VP4 (pfam11492) and CrPV_capsid (pfam08762). E-values were consistently very high for these predicted domains. BPV1 sequences were only assembled from sample 25 (Table 1).

Figure 1 Schematic representation of genome structures of five novel SRV+ with near full length genome sequences.

Genome structures of viruses assembled from BP and BuG transcriptome sequence data with conserved motifs and domains are shown. While tentative coordinates have been assigned, these sequences may not represent the full length viral genomes. Domains are: Calici_CP: Calicivirus coat protein (pfam00916; white box); CrPV_CP: CrPV_capsid (pfam08762; gray); Di: Dicistro_VP4 (pfam11492; lime green); Hel: RNA_helicase (pfam00910; red); Pep: Peptidase_C3 (pfam00548; yellow); RdRP: RNA-dependent RNA polymerase (cd01699, pfam00680; orange); Rh: Rhv_like CP (cd00205, pfam00073; green).

Near full-length small RNA virus sequences assembled from BP Illumina sequencing samples

From the 21 BP samples collected from Kasabong Stream, Kenya in 2013, three near complete small RNA viral genome sequences were assembled. The viruses were named Biomphalaria pfeifferi-associated virus 2 (BPV2: GenBank Accession No. MT108489), Biomphalaria pfeifferi-associated virus 3 (BPV3: GenBank Accession No. MT108490) and Biomphalaria pfeifferi-associated virus 4 (BPV4: GenBank Accession No. MT108491), with genomes of at least 8,491, 8,255 and 8,253 nt, respectively (Fig. 1; Text S1). BPV2 and BPV3 are structurally similar to dicistrovirus-like BPV1, while BPV4 encodes a single ORF, in which non-structural and structural proteins are expressed as a single polypeptide prior to proteolytic cleavage into functional viral proteins. BPV4 is similar in genome structure to Marnaviruses (Fig. 2).

Figure 2 Genome structures of SRV+ discovered from three species of vector snails.

Genome structures are shown for viruses reported here for BP and BuG, and previously for B. glabrata (Adema et al., 2017). BuGV2 and BGV2 have rh_v domains typical of dicistroviruses, but lack any other similarity to known dicistroviruses.

The top BLASTp hits of BPV2 ORF1 and ORF2 predicted polyproteins were the ORF1 and ORF2 polyproteins of Antarctic picorna-like virus 1 (APLV1), which was discovered from Antarctic lakes, but for which the host is unknown (Lopez-Bueno et al., 2015). Protein sequence identities of approximately 31% and 34% were shown between ORF1 and ORF2 predicted polyproteins of BPV2 and APLV1, respectively. Conserved domains identified from the ORF1 polyprotein were RNA-helicase, peptidase_C3 (pfam00548) and RdRP (Fig. 1). In the ORF2 polyprotein, two Rh_v CP domains were identified, with one domain overlapping a Calicivirus CP domain (pfam00916) (Fig 1). The VP4 and a conserved C-terminal CP domain (CrPV_capsid) were not found in the ORF2-encoded protein of BPV2.

The top hits for the ORF1 and ORF2 polyproteins of BPV3 were both from Marine RNA virus-JP B (MRNAVJPB) (Culley, Lang & Suttle, 2007), which is classified as a member of Marnavirus. The sequence identities were ~35% between the corresponding ORF polyproteins of BPV3 and MRNAVJPB. ORF2 of BPV3 encodes 942 aa, which is the longest ORF2 of the dicistrovirus-like sequences discovered from snails. Three CP domains were identified in the ORF2 polyprotein of BPV3, Rhv_like CP domain, VP4, and the CrPV_capsid domain, which also contained conserved sequences of an Rhv_like domain. For all three dicistrovirus-like viruses, translation of BPV2 ORF2 is expected to be mediated by an internal ribosomal entry site (IRES) (Bonning & Miller, 2010).

BPV4 is the only SRV+ identified from BP that encodes a single ORF encoding a polyprotein of 2,685 aa. The genomic structure of this virus is similar to the type virus of genus Marnavirus, Heterosigma akashiwo RNA virus (HaRNAV) (Fig. 2) (Lang, Culley & Suttle, 2004). The N-terminus of the ORF encodes non-structural proteins, while the C-terminus encodes structural proteins. The non-structural proteins of BPV4 have homology to the ORF1 polyprotein of Marine RNA virus SF2, an unclassified dicistro-like virus, and the structural proteins hit the ORF2 polyprotein of Chaetoceros sp. RNA virus 2, an unclassified marine planktonic diatom discistro-like virus (Tomaru et al., 2013a; Tomaru et al., 2013b) with 27% and 29% sequence identities, respectively. The single ORF of BPV4 encodes helicase, RdRP and CP domains (Fig. 1).

Partial SRV+ sequences identified from the BP transcriptome

In addition to the four near complete full-length SRV+ genome sequences described above from BP, 44 putative SRV+ contigs were identified, ranging from 203–3,624 nt (Table S2). Among these partial viral sequences, 23 (52%) were derived from regions encoding non-structural polyproteins, while 21 (48%) contigs hit structural polyprotein sequences. These partial sequences were derived from multiple viruses. The regions from which the sequences were derived were determined by BLAST annotation and are illustrated in Fig. 3. Based on overlap of the structural protein sequences (BPVS 6, 16, 17, 18, 33, 42 43 and 44), the 44 contigs represent at least eight novel SRV+. These eight contigs encode 374–900 aa of the putative partial CP sequences, covering more than 30% of the capsid protein sequences. While 23 contigs hit SRV+ non-structural protein coding regions, only one contig (BPVS1, 1,097 aa) covered 60% of the non-structural polyprotein coding sequence (Fig. 3). Although the protein sequences encoded by the virus-derived RNA fragments hit various SRV+, the majority of hits were from genes encoded by marine RNA viruses (Table S2), similar to the other BPV species described above.

Figure 3 Regions of a generic SRV+ genome with homology to snail-derived virus sequences.

Contigs are labeled BP viral sequence_n (BPVS_1 to BPVS_44) with the number of amino acids encoded shown. The eight distinct but overlapping sequences that hit ORF2 indicative of eight different viruses are shown in bold and boxed.

Other viral sequences

Nine contigs of 206–2,969 nt had homology to several viruses (Table S2), including BPVS52 with homology to a DNA virus infecting the diatom Chaetoceros salsugineum, relatively short sequences with homology to the SRV+ Picalivirus D (Greninger & DeRisi, 2015a) (BPVS51), Puurmala virus (a handavirus of Bunyaviridae; BPVS53) and a plant virus, Currant latent virus (BPVS54). In addition, there were three contigs (BPVS 48, 49 and 50) that hit hypoviruses. While the top hit of BPVS45 is Rosellinia necatrix fusarivirus 1, this contig also hit hypovirus sequences. Hypoviruses are double-stranded RNA (dsRNA) viruses, which are only known to infect Fungi (http://ictvonline.org/virusTaxonomy.asp). Protein sequence identities of the four putative hypoviral sequences to known hypoviruses were 26–50%.

Viral sequences discovered from the B. globosus transcriptome

Fewer viral sequences were assembled from the three sets of BuG sequence data. Only nine contigs derived from viral sequences were identified by BLASTx. From the nine virus-related contigs, one near complete SRV+ genome sequence, named Bulinus globosus-associated virus 1 (BuGV1: GenBank Accession No. MT108492) and a Nora virus-like virus (Bulinus globosus-associated virus 2, BuGV2: Fragment 1 GenBank Accession No. MT108493; Fragment 2, MT108494) were identified. The other putative viral contigs are listed in Table S2.

Similar to BPV1, BPV2 and BPV3, BuGV1 is dicistrovirus-like with 8,528 nt of assembled sequence (Fig. 1). The non-structural ORF1 polyprotein of BuGV1 has homology to MRNAVJPB with 32% sequence identity, while the top hit of the structural ORF2 polyprotein was the ORF2 polyprotein of Marine RNA virus SF-1 (MRNAVSF1) (Greninger & DeRisi, 2015b) with 34% sequence identity. The conserved helicase and RdRP domains were identified from the ORF1 polyprotein, and two Rhv-like and VP4 domains were found in the ORF2 polyprotein (Fig. 1).

In addition to BuGV1, two contigs of 5,278 and 1,529 nt hit an insect Nora virus (VP2, replication related protein), suggesting these two contigs may be derived from the same virus (BuGV2). Nora virus is an SRV+ of Drosophila that encodes four potential ORFs (Habayeb, Ekengren & Hultmark, 2006) (Fig. 4). BLAST annotation showed that the contig of 1,529 nt hit a helicase domain within the Nora virus non-structural proteins, while the contig of 5,278 nt contains two ORFs. The 5′-end of this contig encodes partial sequences of a non-structural protein, while the 3′-half of the contig encodes an ORF of 893 aa (Fig. 4). While this ORF polyprotein did not hit any viral genes, two rhv-like CP domains were identified suggesting that it encodes a putative CP gene. Based on the ORF arrangement, the viral sequences encoded by these two contigs were not derived from a Nora-like virus, but likely from a new type of virus with similarity to dicistroviruses (Fig. 2). BuGV2 currently lacks more than 1 kb of sequence in ORF1.

Figure 4 Putative genome structure of BuGV2 showing structural differences between BuGV2 and Nora virus.

For Nora virus, ORF1 is a highly charged protein of unknown function; ORF2 has helicase, protease and RdRP domains; ORF3 encodes a protein of unknown function; ORF4 encodes structural proteins. The BuGV2 ORF1 has homology to Nora virus ORF2 only. While BuGV2 has two dicistrovirus domains, the encoded protein does not show homology to other viral sequences.

Confirmation of viral sequences

As BPV2-derived sequence was present in all but one of the sequencing datasets used for virus sequence identification (Table 2), we first designed primers (Table S1) to amplify a short fragment (339 nt) to confirm virus sequence presence by RT-PCR. Only two of the 21 BP samples (19 and 20) along with the three BuG samples (22–24), tested negative for the presence of BPV2 (Fig. 5). The absence of BPV2 sequence from BuG as determined by RT-PCR was consistent with the absence of BPV2 sequences in these transcriptomes, and the absence of BPV2 from samples 19 and 20 is consistent with the relatively low number of reads mapped for sequence data based on snails 19 to 21 (Table 2). We then designed primers to generate longer fragments of BPV2, BPV3, BPV4 and BuGV1 (Table S1). BPV2, BPV3 and BPV4 sequences were only amplified from BP snail samples, while BuGV1 sequences were only found in BuG samples (Fig. 5; Fig. S1).

Figure 5 Confirmation of the presence of BPV2, BPV3, BPV4 and BuGV1 sequences in snail RNA samples.

A band of the expected size (339 bp; BPV2 short) for BPV2 was detected on examination of RT-PCR products from 19 of the 21 BP snail samples, but not from BuG (samples 22-24). Longer fragments of BPV2 (BPV2 long), BPV3, BPV4 were detected in some BP samples, but not in BuG samples. BuGV1 was amplified only from BuG samples (See Fig. S1 for uncropped gel images). Lane numbers correspond to sample numbers in Table 1. M, DNA size markers.

Distribution of viral sequences among snail samples

To determine the distribution of virus-derived sequences among the snail samples, we examined the viral sequences by mapping the Illumina sequencing reads (50–100 nt) to the five near complete (Fig. 1) and to seven of the partial virus genome sequences with fragments encoding >800 aa (Table 2). BPV1, which was isolated from a snail collected from Asao, Kenya and sequenced with the 454 platform, was not detected from any of the BP isolated from Kasabong, Kenya. The viral sequences discovered from BuG samples (collected from eastern Kenya) were not detected from the BP samples (collected from western Kenya). Similarly, no reads derived from BuG samples mapped to the viral sequences assembled from the BP samples, except for BPVS6. Only three BuG reads matched the BPVS6 sequence presumably derived from sequence fragments conserved in SRV+. These results suggest that snail-associated virus sequences identified in the current study were specific to both snail species and geography. Among the BP samples collected from Kasabong, the viral sequences were widely distributed across all BP snails. The highest sequence coverage for BP viruses was for BPV2, with 6–263 fold coverage of different regions of the genome. The highest viral sequence coverage was observed from BuG samples, with up to 298-fold sequence coverage for BuGV1 (Table 2).

Phylogenetic analysis of SRV+ identified in snails and related RNA viruses from other organisms

To examine phylogenetic relationships among the viral sequences identified, we used peptide sequences derived from the RdRP domains and CP for analysis. The RdRP sequence is highly conserved among RNA viruses (Koonin, Gorbalenya & Chumakov, 1989; Zanotto et al., 1996), while the viral capsid protein that functions in virus host range is more divergent. Four snail viral genome sequences (Biomphalaria glabrata virus 1, BGV1; Biomphalaria glabrata virus 2, BGV2; Biomphalaria glabrata virus 3, BGV3; and Biomphalaria virus 4, BiV4) identified from RNA sequence data for B. glabrata (Adema et al., 2017; Galinier et al., 2017) were included in the phylogenetic analyses. The virus names from Adema et al., were used in our analysis (based on RdRP sequences: BGV1 is BiV2 from Galinier et al. (2017), BGV2 is BiV1, and BGV3 is BiV3).

Annotation of the snail-derived SRV+ sequences revealed that these viruses are closely related to Marnaviruses (with algal hosts), Bacillarnaviruses (with marine diatom hosts) and insect small RNA viruses (dicistrovirus-like, iflavirus-like and others). There was no similarity to SRV+ infecting marine mammals, e.g., Seal picornalike virus 1 (Kapoor et al., 2008). The RdRP domain is highly conserved in RNA viruses (Koonin et al., 2008). To analyze phylogenetic relationships between the snail-derived SRV+ sequences and similar viruses identified from aquatic environments and insects, the conserved RdRP sequences of selected viruses (based on BLAST annotation results with conserved domain cd01699) were extracted, aligned and used for construction of a phylogenetic tree. This RdRP-based phylogenetic analysis clustered the snail viruses into three groups (Fig. 6): First, BPV1, BPV3, BPV4 and BuGV1 grouped with viruses isolated from diatoms and aquatic environments, and hence these four viruses are unlikely to infect snails. This analysis suggests that the RdRP of BPV3 and BuGV1, and BPV1 and BPV4 are closely related, although the genome of BPV4 encodes a single ORF while that of BPV1 encodes two. RdRP sequences from the other snail-derived viruses separated into two major clusters primarily comprised of insect viruses. For the second group, BPV2 and BPVS1 RdRP sequences grouped with distro-like viruses, along with viruses of marine invertebrates (MCDV, MRTV and TSV) and two marine RNA viruses (APLV1 and LaVUC1) with unknown hosts. Third, based on RdRP sequences, four snail viruses (BGV1, BGV2, BGV3 and BuGV2) together with two unclassified insect viruses (AGV1, APV) and one insect Nora virus (NoV) form a unique clade. It is notable that although closely related with 62.3% RdRP sequence identity, BGV2 and BuGV2 were isolated from different snail genera collected from different geographic locations (BG from South America and BuG from Africa). Although BLAST analysis indicated similarity of some snail virus genome sequences with iflaviruses, none grouped with the iflavirus clade on RdRP analysis. BiV4 did not group into any of the three clusters. The RdRP of this virus is distant from those of the insect viruses or marine viruses used in this analysis.

Figure 6 Phylogenetic relationships among SRV+ RdRP sequences derived from snails, diatoms, marine RNA viruses of unknown hosts and insects.

RdRP protein sequences were aligned and phylogenetic trees constructed by IQ-TREE using maximum likelihood with 10,000 bootstrap cycles. The final trees were drawn using FigTree v1.42. Asterisks indicate diatom viruses. Viruses identified in this study are indicated by boxes. Virus abbreviations (genome accession numbers): ADV: Acheta domesticus virus (DQ112164.1); ABPV: Acute bee paralysis virus (NC_002548.1); AGV1: Aphis glycines virus 1 (MK533146.1); APV: Acyrthosiphon pisum virus (NC_003780.1); APLV1: Antarctic picorna-like virus 1(NC_030232.1); APLV2: Antarctic picorna-like virus 2 (NC_030233.1); APLV3: Antarctic picorna-like virus 3 (NC_030234.1); APLV4: Antarctic picorna-like virus 4 (NC_030235.1); APIV: Antheraea pernyi iflavirus (NC_023483.1); ALPV: Aphid lethal paralysis virus (NC_004365.1); AGRNAV: Asterionellopsis glacialis RNA virus (NC_024489.1); AssRNAV01: Aurantiochytrium single-stranded RNA virus 01 (NC_007522.1); BBV: Brevicoryne brassicae virus (NC_009530.1); BDV: Bat dicistrovirus (MH370347.1); BQCV: Black queen cell virus (NC_003784.1); BMIV: Bombyx mori iflavirus (LC214356.1); ChspRNAV2: Chaetoceros sp. RNA virus 2 (AB639040.1); ChSfrRNAV01: Chaetoceros socialis f. radians RNA virus 01 (NC_012212.1); ChTRNAV01: Chaetoceros tenuissimus RNA virus 01 (NC_038321.1); ChTRNAVT-II: Chaetoceros tenuissimus RNA virus type-II (NC_025889.1); CrPV: Cricket paralysis virus (NC_003924.1); DWV: Deformed wing virus (NC_004830.2); DCPLV: Diaphorina citri picorna-like virus (KT698837.1); DCPV: Dinocampus coccinellae paralysis virus (NC_025835.1); DCV: Drosophila C virus (NC_001834.1); DVvV1: Diabrotica virgifera virgifera virus 1 (KY064174.1); EOPLV: Ectropis obliqua picorna-like virus (NC_005092.1); GDV: Goose dicistrovirus (NC_029052.1); GNV1: Graminella nigrifrons virus 1 (NC_026733.1); FEV1: Formica exsecta virus 1(MH133342.1); FEV2: Formica exsecta virus 2 (NC_023022.1); FSPLV: Fur seal picorna-like virus (NC_035110.1); HaARNAV: Halastavi arva RNA virus (NC_016418.1); HHV: Halyomorpha halys virus (NC_022611.1); HEIV: Heliconius erato iflavirus (NC_024016.1); HeARNAV: Heterosigma akashiwo RNA virus (NC_005281.1); HiPV: Himetobi P virus (NC_003782.1); HCV1: Homalodisca coagulata virus-1 (NC_008029.1); IFV: Infectious flacherie virus (NC_003781.1); IAPV: Israeli acute paralysis virus (NC_009025.1); KaV: Kakugo virus (AB070959.1); KBV: Kashmir bee virus (NC_004807.1); KFV: Kelp fly virus (NC_007619.1); LSHV1: Laodelphax striatella honeydew virus 1 (NC_023627.1); LSPLV2: Laodelphax striatellus picorna-like virus 2 (NC_025788.1); LLV1: Lygus lineolaris virus 1 (NC_038301.1); LDIV1: Lymantria dispar iflavirus 1 (NC_024497.1); MCDV: Mud crab dicistrovirus (NC_014793.1); MRNAVBC-2: Marine RNA virus BC-2 (NC_043542.1); MRNAVBC-3: Marine RNA virus BC-3 (NC_043543.1); MRNAVJP-A: Marine RNA virus JP-A (NC_009757.1); MRNAVJP-B: Marine RNA virus JP-B (NC_009758.1); MRNAPAL156: Marine RNA virus PAL156 (NC_029307.1); MRNAVSF-1: Marine RNA virus SF-1 (NC_043515.1); MRNAVSF-2: Marine RNA virus SF-2 (NC_043518.1); MRNAVSF-3: Marine RNA virus SF-3 (NC_043519.1); MRTV: Macrobrachium rosenbergii Taihu virus (NC_018570.2); NeDVTFN-2012: Nedicistrovirus TFN-2012 (JQ898341.1); NLCV: Nilaparvata lugens C virus (KM270560.1); NLHV1: Nilaparvata lugens honeydew virus 1 (NC_038302.1); NLHV2: Nilaparvata lugens honeydew virus 2 (NC_021566.1); NLHV3: Nilaparvata lugens honeydew virus 3 (NC_021567.1); NV: Nora virus (NC_007919.3); OIIV1: Opsiphanes invirae iflavirus 1(NC_027917.1); PNV: Perina nuda virus (NC_003113.1); PSIV: Plautia stali intestine virus (NC_003779.1); RhSRNAV01: Rhizosolenia setigera RNA virus 01 (NC_018613.1); RhPV: Rhopalosiphum padi virus (NC_001874.1); SaV: Sacbrood virus (NC_002066.1); SssRNAV: SBPV: Slow bee paralysis virus (NC_014137.1); SIV1: Solenopsis invicta virus 1 (NC_006559.1); SEIV1: Spodoptera exigua iflavirus 1 (NC_016405.1); SEIV2: Spodoptera exigua iflavirus 2 (NC_023676.1); TSV: Taura syndrome virus (NC_003005.1); ThPIV1: Thaumetopoea pityocampa iflavirus 1 (NC_026250.1); TMV: Tomato matilda virus (MK517476.1); TrV: Triatoma virus (NC_003783.1); VVARLV: Vespa velutina associated triato-like virus (GGL51738.1). The sequences of KB2009con15, KB2009con28, KB2009con55, KB2009con74, KB2009con88 and KB2010con16 (Culley et al., 2014) were kindly provided by Dr. Alexander I. Culley, Department de Biochimie, de Microbiologie et de Bio-informatique, Universite Laval, Quebec City, Quebec, Canada.

In addition to eight complete small RNA viral CP, five sequence fragments (BPVS6, 16, 17, 33 and 42) encode near full-length putative CP (>90%; Fig. 3) and were included in the phylogenetic analysis based on CP sequences. As with RdRP domains, annotation of the CP sequences also showed protein sequence similarity to several marine viruses and insect RNA viruses (Fig. 7). For phylogenetic trees based on structural proteins, we used the CP or putative CP sequences of selected viruses isolated from aquatic unicellular organisms, marine samples, insect SRV+, and SRV+ discovered from snails (Fig. 7). BPV3, BuGV1, BPVS33 and BPVS42 grouped with marnaviruses, bacillarnaviruses and other unclassified SRV+ isolated from seawater. BPV1, BPV4 and BPVS16 group adjacent to the marine-derived virus groups and separate from BGV2 and BuGV2. Similar to the phylogenetic tree based on RdRP domains (Fig. 6), the CP of BGV2 and BuGV2 group together and based on CP sequences appear to be distant from other viruses isolated from the snail transcriptomes (Fig. 7). BPV2 and BPVS17 grouped with other marine-derived viruses adjacent to the dicistro-like viruses, while BGV1, BGV3, BPVS6 and BiV4 grouped adjacent to the iflavirus group. Interestingly, BPV2 and BPVS17 grouped with Antarctic picorna-like virus 1 (Lopez-Bueno et al., 2015). The RdRP K B2009con15 (Culley et al., 2014) isolated from seawater also grouped with insect related SRV+, suggesting that these viral sequences may be derived from viruses infecting invertebrates. The notable CP sequence variation among the SRV+ isolated from aquatic environments could reflect the complex evolution of SRV+ (Koonin & Dolja, 1993) or recent divergence (Soltis & Soltis, 2003).

Figure 7 Phylogenetic relationships among SRV+ CP sequences derived from snails, diatoms, marine RNA viruses of unknown hosts and insects.

CP protein sequences were aligned, and phylogenetic trees constructed as for Fig. 6. The final trees were drawn using FigTree v1.42. Virus abbreviations as described for Fig. 6. Asterisks indicate diatom viruses. Viruses identified in this study are indicated by boxes.

Taken together, both RdRP and CP sequences suggest that BPV1, BPV3, BPV4 and BuGV1 are aquatic viruses, while BPV2 is dicistro-virus like. While the CP of BGV1 and BGV3 place these viruses as iflavirus-like, this association is unclear from analysis of RdRP sequences.

Discussion

We identified more than 17 diverse novel viral sequences by analyzing transcriptome data for two important snail vectors of schistosomiasis, B. pfeifferi and B. globosus. In so doing, we have characterized one aspect of the complex microbiome encountered by schistosome parasites in their snail vectors (Dheilly et al., 2019), with the potential use of such viruses for biological control for suppression of snail populations. Multiple sequences derived from putative viruses with (+)ssRNA, (−)ssRNA, dsRNA and DNA genomes were identified based on protein sequence annotations (Table S2). However, most viral sequences were derived from novel viruses with (+)ssRNA with more than 12 viruses isolated from BP (four near full-length genomes, and partial sequences from at least eight viruses) and four from BuG (one near full length genome, and three partial sequences). All snails appeared to be associated with multiple SRV+ with five viruses from the present study (BPV2, BPVS1, BPVS6, BVPS17 of BP, BuGV2 of BuG) predicted to replicate in the snail vector based on phylogenetic relatedness to arthropod viruses.

Diverse genome structures evident in snail-derived viruses

The order Picornavirales includes highly divergent viral groups including Caliciviridae, Dicistroviridae, Iflaviridae, Comoviridae, Sequiviridae, Potyviridae and Marnaviridae. This “picornavirus-like superfamily” was classified into six clades based on genomic RNA structures (Koonin et al., 2008). SRV+ discovered from aquatic environments were either dicistro-like (Bacillarnavirus and Labyrnavirus, belonging to Clade 1), or encoded a single polypeptide (Marnavirus, Clade 1), in which the nonstructural proteins are located at the N-terminus, and the structural proteins at the C-terminus (Fig. 2). This type of genome structure differs from those of polioviruses, sequiviruses or insect iflaviruses (genome structure-based Clade 6), which also encode a single ORF, but with the structural and non-structural proteins located at the N- and C-terminus respectively (Fig. 2).

The majority of viruses identified from the two species of snails reported here, along with viruses identified previously from B. glabrata (Adema et al., 2017; Galinier et al., 2017) are similar to dicistroviruses and Marnaviruses (Fig. 2). However, a second type of dicistro-like virus (BGV2 (Adema et al., 2017), Fig. 2) was also identified. The CP of these two closely related viruses has Rh_v but no CrPV domains and show no similarity to other viruses by BLAST analysis, while the ORF encoding nonstructural proteins shows sequence homology to Nora virus that infects Drosophila (Fig. 4). The genome structure of BGV3 (Fig. 2) differs from those of all other viruses found from snails or aquatic environments. The genome structure of BGV3 resembles that of an insect iflavirus. The CP of BGV3 shows partial homology to an insect virus (highest BLAST score with Aphis glycines virus 1; GenBank KF360262.1) and other insect or marine RNA viruses.

SRV+ of insects, snails and unicellular eukaryotes are similar in genome and protein sequence suggesting a common ancestor

The viruses with positive sense, single strand RNA genomes from snails, along with viruses from diatoms and aquatic environments are similar to insect SRV+. This similarity is apparent from the viral genome structure with a single ORF or two ORFs, and from protein sequence similarities. Expression of dicistrovirus ORF2 is mediated by an IRES. Potential IRES sequences were predicted in the intergenic regions (IGR) of BPV1, BPV2, BPV3 and BuGV1 by IRESite (http://www.iresite.org/). Similar to insect SRV+, SRV+ from aquatic environments usually have four capsid subunits (VP1–VP4) that are generated on proteolytic cleavage of the structural polyprotein. The subunit arrangement of the viral structural proteins of snail and diatom infecting SRV+ is similar to that observed in insect infecting dicistroviruses, in that the smallest VP4 is not the first subunit in the structural polyprotein (Fig. 1) (Bonning & Miller, 2010; Liljas et al., 2002). In contrast, in the SRV+ infecting mammals, VP4 is the first CP component (Liljas et al., 2002). Comparison of SRV+ from insects, snails, diatoms and other viruses from aquatic environments suggests that these viruses have a common ancestor.

Not all snail-derived viruses are snail viruses

Virus sequences identified from snail samples reflect virus populations associated with the snail holobiont, i.e., the snail and all associated organisms. The majority of snail-derived virus sequences had similarity to the sequences of unicellular eukaryote-infecting RNA viruses (Marnavirus, Bacillarnavirus, Labyrnaviruses and other unclassified marine RNA viruses), insect SRV+ (dicistroviruses, iflaviruses and others, e.g., Nora virus,) and SRV+ isolated from aquatic arthropods (shrimp and crab). Few BLAST hits were from vertebrate SRV+, e.g., Picalivirus and Calicivirus, commonly found in sewage-derived RNA (Ng et al., 2012). These results suggest that the SRV+ sequences from snails are diverse and likely derived from multiple sources, i.e., from food material, snail parasites or symbionts, or microorganisms passively associated either peripherally or internally with the snail. There was no correlation between schistosome infection or treatment and the presence of any of the viruses identified (Table 2; Buddenborg et al., 2019).

In the past decade, metagenomic analysis of viral sequences from seawater and freshwater revealed previously unknown SRV+ populations in aquatic environments (Culley, Lang & Suttle, 2003; Culley, Lang & Suttle, 2007; Culley et al., 2014; Djikeng et al., 2009; Lopez-Bueno et al., 2015; Mohiuddin & Schellhorn, 2015; Yoshida et al., 2013). However, the majority of the aquatic environment-derived viral sequences were partial, short sequences, and the hosts of these putative viruses are largely unknown. Only a few SRV+ infecting diatoms and other unicellular organisms have been investigated, and three novel (+)ssRNA viral genera, e.g., Marnaviridae, Bacillarnavirus and Labyranvirus in the order Picornavirales have been classified (Lang, Culley & Suttle, 2004; Viruses ICoTo, 2015). The SRV+ sequences discovered from freshwater snails were either close to sequences of diatom-infecting viruses, or of insect-infecting SRV+ and SRV+ with unknown hosts isolated from aquatic environments (Figs. 6, 7: Table S2), suggesting that the SRV+ from freshwater snails may include viruses that infect both unicellular organisms and aquatic invertebrates. Indeed, phylogenetic trees based on RdRP and CP, consistently grouped BPV1, BPV3, BPV4 and BuGV1 with diatom and marine RNA viruses, and BPV2 with invertebrate/dicistrovirus-like viruses. The CP (but not RdRP sequences) grouped BGV1 and BGV3 with iflaviruses (Figs. 6, 7).

Although empirical tests are required to establish which viruses infect snails, based on sequence similarity and phylogenetic relationships, we conclude that BuGV2 and BPV2 along with viruses from which genome fragments BPVS1, BPVS6 and BPVS17 were derived, are likely to be snail-infecting viruses. This work also supports the likelihood that BGV1, BGV2, BGV3 and BiV4 are snail-infecting viruses. Further investigation is needed for confirmation of a subset of these viruses, identification of their respective hosts, and assessment of chronic versus acute pathology for those that replicate in the snail. The Bge cell line derived from embryos of the schistosome-transmitting snail Biomphalaria glabrata (Rinaldi et al., 2015) will provide an invaluable tool for further study of molluscan virology, potentially allowing for in vitro propagation and analysis of the putative snail viruses described herein.

Supplemental Information

Supplemental Information 1 Confirmation of the presence of BPV2, BPV3, BPV4 and BuGV1 sequences in snail RNA samples (uncropped gel images showing RT-PCR products).

Long fragments of BPV2, BPV3, BPV4 sequences were detected in some BP samples, but not in BuG samples (22 and 24). BuGV1 amplification product was amplified only from BuG samples. Lane numbers correspond to sample numbers in Table 1. M, DNA size markers.

Click here for additional data file.

Supplemental Information 2 Primers used for RT-PCR amplification of viral sequences from four viruses in two snail species.

Click here for additional data file.

Supplemental Information 3 List of virus-derived partial sequences isolated from BP and BuG with details of similarity to other viruses.

Fifty-four partial viral sequences were identified from BP, and three from BuG. Sequence lengths and details of virus proteins with highest similarity based on BLAST analysis are shown. Accession numbers for all sequences are provided.

Click here for additional data file.

Supplemental Information 4 Sequence data.

Click here for additional data file.

The authors thank Dr. John VanDyk for IT support at Iowa State University, and Martin Mutuku from KEMRI for assistance with collection of snails. Technical assistance was provided by the University of New Mexico Molecular Biology Facility. The content for this paper is solely the responsibility of the authors and does not necessarily represent the official views of the National Institutes of Health.

Additional Information and Declarations

Competing Interests

Author Contributions

Field Study Permissions

DNA Deposition

Data Availability

The authors declare that they have no competing interests.

Sijun Liu conceived and designed the experiments, performed the experiments, analyzed the data, prepared figures and/or tables, authored or reviewed drafts of the paper, and approved the final draft.

Si-Ming Zhang performed the experiments, prepared figures and/or tables, authored or reviewed drafts of the paper, and approved the final draft.

Sarah K. Buddenborg performed the experiments, authored or reviewed drafts of the paper, and approved the final draft.

Eric S. Loker conceived and designed the experiments, authored or reviewed drafts of the paper, coordinated with appropriate authorities to allow for this work to be done, and approved the final draft.

Bryony C. Bonning conceived and designed the experiments, analyzed the data, prepared figures and/or tables, authored or reviewed drafts of the paper, and approved the final draft.

The following information was supplied relating to field study approvals (i.e., approving body and any reference numbers):

This project was undertaken with approval of Kenya’s National Commission for Science, Technology and Innovation (permit number NACOSTI/P/15/9609/4270), Kenya Wildlife Service (KWS-0045-03-21 and export permit 0004754), and National Environment Management Authority (NEMA/AGR/46/2014 and NEMA/AGR/149/2021).

The following information was supplied regarding the deposition of DNA sequences:

The five full length and 57 partial virus genome sequences are available in Table S2. The accession numbers are MT108488 to MT108548.

The following information was supplied regarding data availability:

Data are available at NCBI BioProject under PRJNA383396 and PRJNA451210.

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
