# Peer review of "Virus-derived sequences from the transcriptomes of two snail vectors of schistosomiasis, Biomphalaria pfeifferi and Bulinus globosus from Kenya"

_PeerJ, doi:10.7717/peerj.12290_

## Round 0.1 · original submission · Minor Revisions

We received three very comprehensive reviews that I feel are quite constructive and should help improve the final manuscript. Note that (especially under the current COVID circumstances), no further wet lab work is required (as mentioned, but not recommended by Reviewer 3) but it would be worth exploring if some of the partial sequences can be extended. The remainder of the comments/concerns appear to be fairly straightforward to address.

Reviewer 1 ·

Basic reporting

This study describes viruses detected in two biomedically relevant snail species known to vector human-infecting schistosomes that cause debilitating diseases affecting millions of people. Detected viruses were dominated by positive-sense single-stranded RNA (+ssRNA) viruses and represent a diversity of viruses potentially infecting the snails themselves and other organisms associated with the snail holobiont. The viruses were detected through transcriptome analysis and authors confirmed the presence of some of the viruses in snail samples through PCR assays. Authors’ conclusions were not overblown and the generated data adds to the increasing viral diversity reported from invertebrates. The study provides information regarding viruses that could be further explored for biocontrol strategies of snails harboring and transmitting parasitic schistosomes. Is also important that we gain a better understanding of the microbiome of these snails and how it might affect schistosome infection/transmission. This study provides important baseline information regarding viruses associated with snail vectors. See general comments for the author for minor comments regarding basic reporting.

Experimental design

No major concerns, see general comments for the author

Validity of the findings

No major concerns, see general comments for the author

Additional comments

General comment:
Authors need to clarify/highlight that they used the same samples used for another study investigating snail responses to schistosome infection (Buddenborg et al 2017). Therefore, the sample set includes control snails (uninfected) and snails representing various infection treatments. This could be done briefly and authors should discuss if there were any trends (or not) in virus positive samples. For example, were viruses detected in both infected and uninfected snails?

Specific comments:
Introduction:
Line 57. Are there specific examples of viruses used as biocontrol agents of snails or is this a general comment/assumption? Please clarify and refer to previous work (either in snails or other systems).
Lines 64-66. Note that this is a general trend for eukaryotes as a whole given that +ssRNA viruses dominate the eukaryotic virome.
Line 77. Did the 1,300 transcripts represent +ssRNA viruses?

Methods:
Lines 89 – 94. Authors need to specify here that the numbers in parenthesis refer to sample numbers shown in Table 1.
Line 121. What was the quality score threshold used?
Line 124. This is where the reader would benefit from some background about the original study since suddenly authors refer to triplicate snails (triplicate of what?). In the previous paragraph authors stated that there were 3 sequencing replicates per sample but that is not the same as triplicate snails.
Lines 131-132. Authors need to specify thresholds used. For example, what was the minimum contig coverage to count a sequence as present (e.g., half the contig needed to be covered?) and what was the minimum sequence identity needed to map a given read? This is important because if most reads are only mapping to a given region they might represent related viruses that have a conserved region but are not the same viral species
Line 136. Authors need to specify what is in the local viral protein database (e.g., viral protein sequences from RefSeq? If so, which release?)
Line 137. Which version of the NCBI nr database was used?
Line 145. cDNA was synthesized using random hexamers?
Lines 147-149. Specify fragment sizes considered short (<1 kb?) or long (>1kb?).

Results
Lines 171 – 173. Given that EVEs are not only retroviral in nature, how did authors check for EVEs? Did they search snail genome sequences (if available) for detected viruses? Or did they simply eliminate all retroviruses from their data?
Lines 173 – 174. Authors should consider highlighting the 5 near-complete genomes on Table 2. Currently, is hard to figure out the 5 genomes until you get to the last footnote
Line 207. Refer/rephrase to ‘genome structure’ or ‘genome organization’ here
Line 242. Authors should consider a better way of highlighting the contigs of interest in Figure 3. Currently the 8 contigs do not stand out with the bold font alone.
Line 251. Were authors expecting DNA viruses in their data?

Authors should consider switching the order of the ‘confirmation of viral sequences’ and ‘viral sequences discovered from the B. globosus transcriptome’ sections. Currently, the former section discusses PCR for BG viruses that have not been discussed until you get to the next section.

Line 274. Do authors think this outcome simply reflects sampling effort? (only 3 BuG samples)
Line 281. Is BuGV2 not listed in Figure 1 because is not a near-complete genome? This is a bit confusing since it is shown on Figure 2 as one continuous contig. Should ORF1 of BuGV2 in figure 2 be partial (somehow showing an incomplete ORF) assuming the schematic is showing the 5,278 bp contig? I noticed this is stated in line 296 but it should be shown on the figure.
Line 286-296. Did authors attempt to bridge the gap between the two contigs through PCR? Seems like they predicted a 1500nt gap, which in theory should be easily amplified (Fig 5). This would support that both contigs represent the same virus. I’m not suggesting that authors do more lab work, but it would be informative to know if attempts were made to bridge the gap.
Lines 293 – 294. Based on the genome of structure of?
Lines 325-340. I believe trees shown in figures 8 and 9 are more informative than trees shown in figures 6 and 7, which seem redundant and less reliable (less information for tree building). Authors should consider eliminating current trees shown on figures 6 and 7.
Figures 8 and 9. Why authors did not distinguish between diatom (known host) and marine viruses (unknown host) on the trees?
Line 360. Do authors mean insect Nora virus?
Lines 381 – 382. How did authors come to this conclusion? The snails were sampled from freshwater. Moreover, most virome studies have been done in marine systems and there is not much data from freshwater environments. The snail viral sequences are most closely related to marine viruses, which may have to do with a database bias, and it does not mean that the identified viruses are marine (especially when considering the limited amino acid level sequence similarities).

Discussion:
Line 394. This sentence needs more information. These viruses were predicted to replicate in the snail based on their phylogenetic association with?
Lines 395 – 415. These paragraphs seem redundant with the last section of the discussion. Therefore, authors should consider merging this text with the last section.
Line 422-426. Note that Koonin et al 2008 classified iflaviruses (alongside discistroviruses and marnaviruses) under Clade 1 (which is composed of 3 subclades) based on RdRp phylogeny. Please confirm and edit accordingly.
Lines 431-437. This section reads more like results. What is the importance/function of the Rh_V and CrPV domain (if known)? Authors mention BGV2 and BuGV2 represent a second type of dicistro-like virus but these viruses seem to have a different genome organization (Figure 2)
Line 432. ‘show no similarities to other viruses’ through BLAST?
Line 435. Rephrase to ‘genome structure of BGV3’
Line 437. Why highlight Aphis glycines virus 1 but not the ‘other insect viruses’? Is the main point here that BGV2, BGV3 and BuGV2 are most similar to insect viruses?
Line 439-454. How do these findings compare to analysis/phylogeny done by Koonin et al 2008, where all these groups were classified under the same clade?
Line 466-467. Could this be due to recombination?
Lines 474-477. Sounds promising!

Reviewer 2 ·

Basic reporting

1. The report by Liu S. et al 'Virus-derived sequences...Kenya" is written well, the article is clear, and its scientific English is informative and clear in its meaning and intent.

2. Generally, the report is referenced appropriately. However, the inclusion of the citation of Dheilly NM, Martínez Martínez J, Rosario K, Brindley PJ, Fichorova RN, Kaye JZ, Kohl KD, Knoll LJ, Lukeš J, Perkins SL, Poulin R, Schriml L, Thompson LR. Parasite microbiome project: Grand challenges PLoS Pathog. 2019 Oct 10;15(10):e1008028. doi: 10.1371/journal.ppat.1008028 is recommended, along with comments on the information in Dheilly et al 2019 and the authors' current findings and Discussion.

3. Professional article structure, with generally sufficient field background and context provided. Raw data evidently shared via BioProject ID PRJNA383396.

4 The report is self-contained unit with results relevant to the hypothesis that snail intermediate hosts of schistosomes are host to snail specific viruses and with associated environmental viromes.

Experimental design

Experimental designs and analysis of findings look adequate and appropriate.

Validity of the findings

Novel viruses identified in Biomphalaria pfefferi and Bulinus globosus.

Conclusions stated well.

Additional comments

Suggestions to enhance the report.

1. However, the inclusion of the citation of Dheilly NM, Martínez Martínez J, Rosario K, Brindley PJ, Fichorova RN, Kaye JZ, Kohl KD, Knoll LJ, Lukeš J, Perkins SL, Poulin R, Schriml L, Thompson LR. Parasite microbiome project: Grand challenges PLoS Pathog. 2019 Oct 10;15(10):e1008028. doi: 10.1371/journal.ppat.1008028 is recommended, along with comments on the information in Dheilly et al 2019 and the authors' current findings and Discussion.

2. L172. Provide the EVE sequences in the Supporting information as a Supplementary Table.

·

Basic reporting

The MS is well written, clear and unambiguous.

References are sufficient (although in general, many references of non primary results are somewhat old and could be updated). Snail virus references are scarce because gastropod virology literature is rare. In this context the results are novel and relevant given the medical importance of these specific snails.

The article is well structured, I provide below some suggestions to improve figures. The raw data has been shared.

Experimental design

The conducted research is within the aims and scope of the journal. The experimental design is well defined, research questions and knowledge gaps are addressed rigorously using previously validated methods, which are described in sufficient detail. I provide a few comments below to improve the M&M section.

Validity of the findings

all underlying data has been provided. No statistical analyses are included (not aplicable), conclusions are well stated.

Additional comments

Liu et al present an interesting work and valuable findings from the transcriptomes of two snail vectors of schistosomiasis, Biomphalaria pfeifferi and Bulinus globosus from Kenya. Authors have re-assessed previously reported RNAseq datasets in order to identify their RNA virus component. To this end, authors employed a pipeline for virus discovery, which they have previously reported and validated on NGS data from insects. They provide virus like sequences, a few coding complete and several partial sequences, which appear to correspond to novel virus species associated to these snails. Given that the authors provide phylogenetic analysis, where some of these viruses cluster with four Biomphalaria glabrata viruses reported previously, there is plausible indirect evidence that these virus sequences could correspond to putative bona fide viruses of snails.

In addition, authors identified viral sequences, which they describe as being from the entire snail holobiont, including symbionts, ingested material and organisms passively associated with the snails.

Moreover, authors have deposited the virus sequences in NCBI GenBank and raw data at NCBI SRA, reinforcing availability of reported materials, transparency, and reproducibility or their results.

The manuscript is well written and presented. I have only some suggestions for the authors and a few questions for clarification.

Main concerns:
-This reviewer feels that with little effort authors might be able to sort out many of the partial virus sequences described. I am not suggesting Sanger sequencing the gaps to complete genomes, which would be optimal but requires further wet lab. Just some additional work on the available sequences and raw data. There is a lot of redundancy and some overlapping in the partial sequences reported, which as is, do not help much in presenting a proper glimpse of these snail virome. For instance: “44 contigs of “at least” 8 viruses” as a back of the envelope calculation is not very helpful to describe virus diversity.
As an example, I provide attached to this review a nearly complete genome with a few Ns (ca. 9.25 kb) of a hypo-like virus, which was crafted just arranging BPVS8, BPVS9, BPVS45, BPSV48, BPVS49, BPSV50 and a few contigs of your reported TSA (GFMW).

- small RNA viruses (SRA): Authors keep refereeing to the detected ca. 8kb genome viruses as “small”, however, to my knowledge that would be the average size for a RNA virus genome (given the great abundance of picornavirales members linked to invertebrates, or plant potyviruses). Unless you find some literature to contrast this I usually consider small RNA viruses members of mitovirus (ca. 2.8 kb), other narnaviruses or amalgaviruses (ca. 3.4kb), for instance, and large genome RNA viruses would be nidoviruses (ca. 13-41 kb).

-many detected viruses in the snail dataset appear to cluster close to diatoms or marine organisms of unknown hosts. One obvious reason could be the very few snails viruses available in the literature redounding in viral dark matter and a general lack of “snail viruses clades” which is in line with the very low sequence identity of the detected viruses and their closest hits (ca. 30% id). Nevertheless, in a context where most detected viruses closest counterparts correspond to non-snail viruses, at least for those, the authors could consider the possibility of shifting their proposed names to a more conservative one which won’t be affected if further studies confirm that their proper host is a snail holobiont species and not the snail itself. This reviewer suggest using for instance, “Biomphalaria pfeifferi associated virus 1” or something in that line until more direct evidence confirms that these viruses do infect the snail.

It is worth mention that most of the reported nearly complete RNA viruses, as suggested by the authors, appear not to be infecting the snails (BPV1, BPV3, BPV4 and BuGV1).

-I wonder if a very simple assessment of the primary sequence of these novel viruses could provide some clues to host assignment. For instance if you check codon usage of BPV1-BPV4 would you get something similar to gastropods codon usage??

-Here you describe many viruses probably linked to the snail holobiome. Could you try a rapid examination of the putative holobiome using the very same data? For instance running a quick rRNA assessment of each of your RNAseq libraries may show some common and rare symbionts (contaminants?) co-purified in your samples.

-consider discussing the putative virus RNA component as a share of the RNA data and some insights about potential virus abundance in the samples. E.g., what percentage of total reads correspond to virus derived reads in each sample? Which virus appear to be present at higher RNA levels in the studied samples? Etc.

Minor comments

-Introduction: Please consider defining trematodes and any literature available regarding schistosoma infecting viruses.

-Please make sure that the snail virus sequences are released by GenBank. Consider posting them as supplementary data of this article in case the seqs are not available at NCBI promptly.

-As indicated in Table 1 the RNAseq datasets used here from Buddenborg et al. 2017 correspond not only to different organisms but also to different treatments/conditions. Please consider discussing the potential effect of these treatments to the RNA levels of some of the detected viruses in terms of raw mapping differences.

Minor editorial comments (please do not hesitate to ignore all these)

Abstract:
-line 30: “using viral RNA”… add using predicted viral RNA…

Mat&Met:

-line 128: “Virus-derived contigs that encoded >800 aa…” (many (most) of the reported partial virus sequences encode proteins well below that size threshold, please clarify which sequences and using which criteria were selected for analyses and abundance.)

-131-132: “MAQ v0.5.0 (http://maq.sourceforge.net/maq-man.shtml) was used for mapping of reads to target assembled sequences.” Please provide parameters/commands used to implement MAQ.

-line 136: “against a local viral protein database” please define or provide this database in order for results to be reproducible for readers.

-line137: “Contigs that hit viral proteins were extracted” using which identity, and/or e-value and/or coverage threshold?

-line 139: how did you manually check that? Using which tools? Maybe bioedit mentioned below?

-line 142-151: could you clarify if RT-PCR confirmation of viral presence was conducted in the very same RNA which was subjected to HTS sequencing or to independent RNA extractions?

-line 154-155: “conserved domains ….were identified by blast”, this reviewer did not have access to the fully annotated sequences so could not contrast your predictions but may I suggest authors to check their predictions with additional (and perhaps more reliable) tools such as HHpred https://toolkit.tuebingen.mpg.de/tools/hhpred or NCBI CDD search https://www.ncbi.nlm.nih.gov/Structure/cdd/wrpsb.cgi ?

-line 159: “The domain sequence was aligned using MAFFT” please provide parameters (algorithm, scoring matrix, etc).

Results

-line 172: “endogenous viral element (EVE) sequences were not included in this analysis” how did you detect (and exclude) potential EVEs?

-line 187, line 208 and elsewhere: “ORF1 of BPV1 showed 30% protein…” please here and elsewhere clarify that the reported identity derives of comparing the ORF1 encoded predicted (poly)proteins.

-line 192-196 and elsewhere: could the authors provide some support of the predicted domains such as e-value?

-line 203 and elsewhere: “with genomes of 8,491, 8,255 and 8,253 nt” without RACE or other experimental confirmation it is not possible to predict the exact genome size only by HTS data assembly.

-Please provide sequence identity values of BPV1-BPV4 nt and predicted domains among them, which supports that the sequences correspond to distinct virus species. Also provide ICTV species demarcation criteria for picornavirales in general.

-line 360: “one insect norovirus” replace norovirus with Nora virus

-line 364: “could potentially be an ancestral virus” please rephrase, a contemporary virus should not be described as ancestral.

Discussion

-Line 388: “(-)ssRNA, dsRNA and DNA genomes” the (-)ssRNA and DNA hits on table 2 appear to be more contamination thank snail viruses.

-line 393 “predicted to replicate” I would rephrase, to my understanding there are no direct evidence of replication in this work for any virus.

-Line 456: “Not all snail-derived viruses are snail viruses” please consider discussing contamination as a potential source of putative RNA viruses.

-Table S2: row 59 The 893 aa predicted product of BuG_virus2 returns a significant hit (e-value = 1e-71) with the structural protein of Biomphalaria virus 1 (NC_032804.1).

Figure comments

Figure 1. blue (ORF1) and green (ORF2) is missing in the legend. Please provide coordinates of orfs and/or domains. You may add some symbol indicating a Poly(A) at 3’ end termini. Perhaps figure 1 and 2 could be integrated into one with a and b panels.

Figure 3. There aren´t much info here. maybe move this one to supplementary material?

Figure 6 and 7. Legends appear to be incomplete, which viruses are those? What’s the scale bar representing here? What are the numbers on the nodes?. In addition, not sure what is the general idea to generate these two trees without all available viruses as in figure 8 and 9. If the idea here is to highlight each of those corresponds to a different species, then genetic distances (nt and aa) as a plot would be more useful.

Figure 8 and 9. Please add an asterisk to highlight each of the viruses detected in this study. Also both trees are missing scale bars. The node labels are illegible. How /was this tree rooted? How did you selected which virus sequences to incorporate in these trees?

---

## Round 0.2 · accepted · Accept

Thank you for your thoughtful responses to all of the reviewers' comments and suggestions.